# Irreversible temperature gating in trpv1 sheds light on channel activation

Ana Sánchez-Moreno[1], Eduardo Guevara-Hernández[1,2],
Ricardo Contreras-Cervera[2], Gisela Rangel-Yescas[1], Ernesto Ladrón-de-Guevara[1],
Tamara Rosenbaum[2], León D Islas[1]*

[1]Departamento de Fisiología, Facultad de Medicina, México City, México; [2]Instituto de Fisiología Celular, México City, México

**Abstract** Temperature-activated TRP channels or thermoTRPs are among the only proteins that can directly convert temperature changes into changes in channel open probability. In spite of a wealth of functional and structural information, the mechanism of temperature activation remains unknown. We have carefully characterized the repeated activation of TRPV1 by thermal stimuli and discovered a previously unknown inactivation process, which is irreversible. We propose that this form of gating in TRPV1 channels is a consequence of the heat absorption process that leads to channel opening.
DOI: https://doi.org/10.7554/eLife.36372.001

## Introduction

Thermo TRPs are unique in that they can be activated by changes in temperature alone (*Latorre et al., 2009*). In particular, TRPV1 channels respond to temperature increases with a large surge in current, produced both by an increment in the open probability and in the single channel-conductance, with the open probability being more temperature dependent. The mechanism by which absorbed heat is converted into a conformational change triggering channel opening remains largely unknown (*Castillo et al., 2018*).

A large positive enthalpy change in the order of tens of kcal·mol$^{-1}$ is required for TRPV1 activation (*Yao et al., 2010a*). This heat dependence is mostly associated with channel opening while the closing transition is not highly temperature-dependent (*Yao et al., 2010a*; *Liu et al., 2003*). This is the opposite in cold-activated channels, like TRPM8 (*Raddatz et al., 2014*).

Along with these thermodynamic constrains, TRPV1 behaves as an allosteric protein. Allosteric activation is evidenced by multiple open and closed states (*Liu et al., 2003*) and is also observable as a coupling between distinct activation modes when channels are activated by more than one stimulus (*Ahern et al., 2005*; *Jara-Oseguera and Islas, 2013*; *Brauchi et al., 2004*; *Jara-Oseguera et al., 2016*; *Cao et al., 2014*).

In contrast with the voltage sensor of voltage-gated channels, a 'temperature sensor' has not been found and it has been suggested that temperature might affect several regions of the channel protein at once, explaining why so many regions have been implicated in temperature gating in thermoTRP channels (*Cui et al., 2012*; *Grandl et al., 2008*; *Yao et al., 2010b*; *Yao et al., 2011*; *Brauchi et al., 2006*; *Wang et al., 2013*). A possible mechanism for channel gating by temperature invokes a large difference in heat capacity between the closed and open conformations (*Clapham and Miller, 2011*; *Chowdhury et al., 2014*). One possible way of realizing this mechanism is a change in solvation of hydrophobic or hydrophilic regions or residues to a hydrophilic or hydrophobic environment, respectively.

In this report, we investigated the repeated activation of TRPV1 channels by temperature. We find that activation is coupled to a previously uncharacterized temperature-dependent inactivation

**\*For correspondence:**
leon.islas@gmail.com

process and show that this transition is irreversible. Our data are suggestive of a partial or complete unfolding of one or more regions of the channel protein during heat absorption.

## Results and discussion

### Activation of TRPV1 channels by temperature ramps

*Figure 1A* shows the time course of the current response to a typical temperature ramp. Since these temperature ramps change at a rate of ~15 °C/s and activation of TRPV1 by fast temperature jumps has a time constant in the order of ms (*Yao et al., 2009*), our ramp measurements can be considered to be in quasi steady-state. Temperature changes in an almost completely symmetrical fashion, increasing and decreasing at almost the same rate. The resulting TRPV1 temperature-activated current is shown in the same figure at two different voltages. TRPV1-mediated currents can be activated at both negative and positive potentials. Channel activity can be observed even at 20°C in multichannel patches, and it visibly increases around 40°C (*Figure 1B*). Current activates and deactivates steeply, but in spite of the symmetric heat stimulus provided by the ramp, the current response is not symmetrical (*Figure 1C*). This can be clearly seen when current is plotted as a function of ramp temperature (*Figure 1D*). It is apparent that the current during temperature decrease is less steep than the up-ramp current, so current activation-deactivation is accompanied by a high degree of hysteresis. Hysteresis is not a consequence of the speed of the ramp, since a slower ramp (7 °C/s) produces the same result. Also, activation by slower ramps proceeds with a high enthalpy, not significantly different from activation by the faster ramp (*Figure 1—figure supplement 1*).

Transformation of the up and down currents to a van't Hoff plot, allowed for the estimation of the apparent enthalpic change ($\Delta H$) involved in opening and closing of the channels, as the slope of an exponential function fitted to the steep part of the curves. As mentioned before, deactivation proceeds with a smaller associated enthalpy (*Figure 1E and F*).

### TRPV1 inactivates with repetitive activation

A key novel observation is that the magnitude of the TRPV1 temperature-activated current is reduced with each subsequent ramp. Strikingly, after several activation episodes, currents almost completely disappear (*Figure 2A*). The reduction of current becomes more pronounced at higher temperatures. When the ramp reaches temperatures near 55°C, current decay is visible even before the ramp gets to its maximum temperature (*Figure 2B*). The decay of current as a function of time is temperature-dependent. Current loss when activating by ramps that reach moderate temperatures (41–43°C) proceeds more slowly than when channels are activated by ramps above 50°C (*Figure 2C*). Current loss upon heating also occurs when the patch is held at a negative voltage of −60 mV (*Figure 2D*). Activation of TRPV1 is more steeply dependent on temperature at negative voltages, which is reflected in the enthalpy of activation being significantly higher at −60 mV than at 60 mV (*Figure 2E*), as expected from its allosteric coupling (*Jara-Oseguera et al., 2016*). Concurrently, current loss at negative voltage proceeds faster than at positive voltages for the same temperature (*Figure 2F*). Current loss is not caused by patch excision, since it can be observed in whole-cell and outside-out recordings, with similar characteristics (*Figure 2—figure supplement 1*). This suggests that the mechanism of current loss is a property of the TRPV1 polypeptide and is not caused by the loss of a stabilizing factor upon patch excision, as described for other TRP channels. In a previous study (*Liu and Qin, 2016*), it was observed that TRPV1 activation, after challenging the channel with two high temperature pulses was stable. The present study shows that inactivation is observed when a series, more than two, of high temperature pulses are applied.

An important observation is the fact that the reduction of current amplitude resulting from repetitive stimulation is accompanied by a dramatic change in the sensitivity of channels to temperature. This sensitivity is mainly determined by the apparent activation $\Delta H$. The response of TRPV1 to the first ramp results in a very steep activation curve. Subsequent activation curves show reduced slopes, indicating a reduction of the $\Delta H$ associated with opening. After five ramps, reaching a maximum temperature of 48.7°C, the $\Delta H$ of activation is reduced to near 10 kcal·mol$^{-1}$, or a $Q_{10}$ value of 1–2, which is similar to diffusion processes (*Sidell and Hazel, 1987*) (*Figure 2—figure supplement 2*).

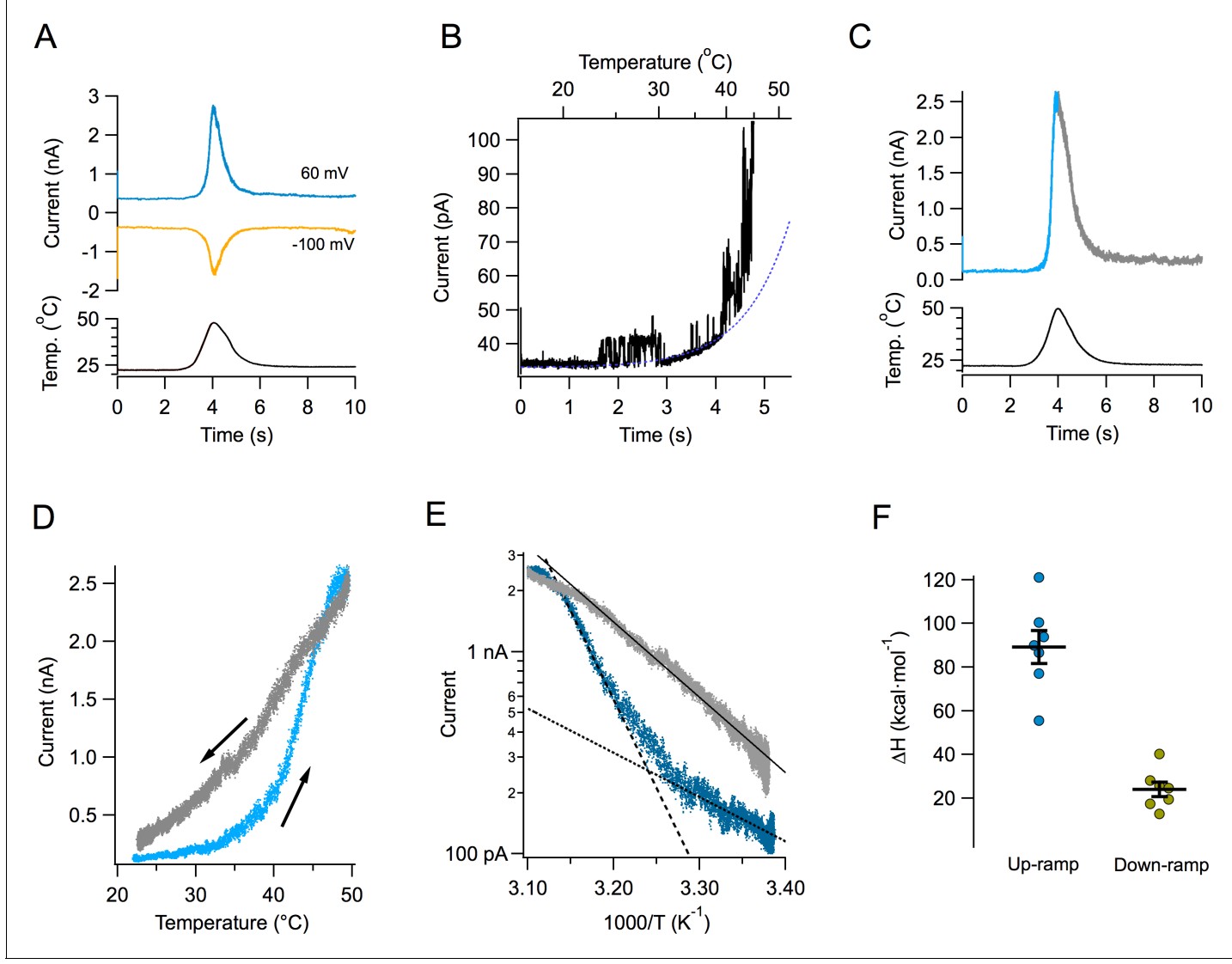

**Figure 1.** Activation of TRPV1 current by temperature ramps. (A) TRPV1-mediated Na$^+$ currents in an inside-out patch, activated by a temperature ramp while the voltage was held constant at the indicated value. The ramp had a total duration of 4 s (2 up and 2 down) and is shown below the currents. The maximum temperature reached was 47.8°C. (B) Single channels can be observed in a patch containing hundreds of TRPV1 channels. In this patch, the detectable increase in open probability happens around 40°C. (C) A different patch showing the up and down regions of the current in response to the temperature ramp in different color. Notice the asymmetry in the kinetics of the current. (D) The current in panel (C) is plotted as a function of the temperature ramp. The arrows indicate the direction of temperature increase (blue) or decrease (gray). It is evident that the paths followed by activation and deactivation are different. (E) Van't Hoff representation of the current during the up section of the ramp (blue) and the down section (grey) as a function of inverse temperature. The threshold of current activation is determined from the intersection of the two exponential functions representing the leak current (black dotted line, enthalpy 10 kcal· mol$^{-1}$) and the TRPV1 current (black dashed line, enthalpy 40 kcal·mol$^{-1}$). The continuous black line is an exponential fit to the deactivating (down ramp, enthalpy 17 kcal·mol$^{-1}$) current. (F) Activation properties of TRPV1 by heat. The enthalpy was obtained from seven patches.

DOI: https://doi.org/10.7554/eLife.36372.002

The following source data and figure supplement are available for figure 1:

**Source data 1.** Enthalpy values for *Figure 1F*.
DOI: https://doi.org/10.7554/eLife.36372.004

**Figure supplement 1.** Hysteresis is present and identical in longer temperature ramps.
DOI: https://doi.org/10.7554/eLife.36372.003

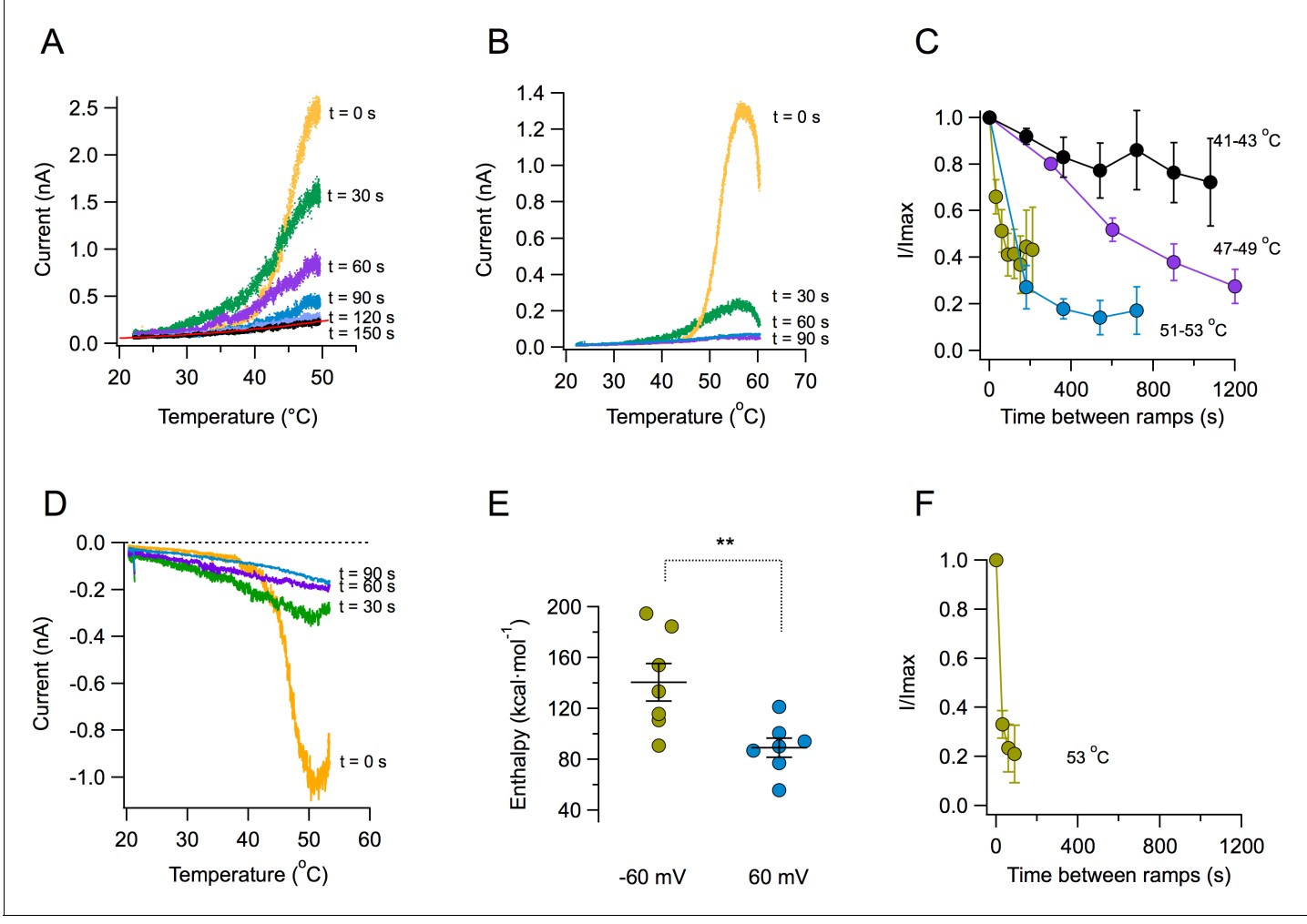

**Figure 2.** Loss of current with repetitive activation in TRPV1. (**A**) Activation (up) part of the response to a temperature ramp in an inside-out patch expressing TRPV1 channels. The membrane potential was 60 mV. The maximum temperature attained was 49.7°C. Ramps of total (up and down) duration of 4 s were applied every 30 s. Note the marked reduction of peak current as well as the decreased slope of activation of every subsequent ramp-activated current. The trace at 150 s is essentially the response of the patch, since all TRPV1 current has been lost. This trace is fitted to an exponential dependence of temperature with an associated enthalpy of 14 kcal·mol$^{-1}$. (**B**) Response of a different patch to a ramp of higher temperature (peak temperature 59°C). The loss of current is evident even during the up ramp response before the ramp has reached its maximum temperature. The response to subsequent ramps is much more diminished at this higher temperature. (**C**) Time course of current loss as a function of different maximum ramp temperatures. Maximum temperatures attained during the activation ramps are: Black circles, 41–43°C, purple circles, 47–49°C, blue circles, 51–53°C, lemon circles, 51–53°C, with a shorter interval between ramps. (**D**) Current loss also happens at negative voltages. Response of a patch held at −60 mV to repetitive ramps applied every 30 s. The maximum temperature attained by the ramp was 52.9°C. (**E**) Activation is more temperature-dependent at negative voltages. Enthalpy of activation estimated from the slope of activation of several independent patches at −60 and 60 mV. The mean enthalpy is different at the two voltages (p<0.001). (**F**) Current loss proceeds faster at negative voltages. Currents at −60 mV decay faster than currents at 60 mV at the same temperature (~53°C, compare with the lemon symbols in part C).
DOI: https://doi.org/10.7554/eLife.36372.005

The following source data and figure supplements are available for figure 2:

**Source data 1.** Raw data for *Figures 2C, E and F*.
DOI: https://doi.org/10.7554/eLife.36372.008

**Figure supplement 1.** Current loss with repetitive stimulation is observed in whole-cell and outside-out recordings.
DOI: https://doi.org/10.7554/eLife.36372.006

**Figure supplement 2.** Inactivation by heat is accompanied by a loss of heat sensitivity.
DOI: https://doi.org/10.7554/eLife.36372.007

## Thermal inactivation of TRPV1 is irreversible

In order to further characterize current loss with repetitive thermal stimuli, currents were first inactivated by more than 50% and then we attempted to reactivate them after a recovery period. Surprisingly, current could not be elicited after the recovery period. Lack of recovery is observed when current is inactivated by application of several ramps at moderate temperature (*Figure 3A and C*) or during a single ramp at high temperature (*Figure 3D and F*). Since current loss is highly temperature dependent (*Figure 2*), we investigated if lowering temperature could help in the recovery process. Again, current loss was irreversible even if the temperature was lowered to ~10°C during the recovery period (*Figure 3B,C,E and F*). Finally, extending the recovery period from 2 to 20 min failed to produce any recuperation of current, regardless of the membrane potential value during recovery (0 or −60 mV, *Figure 3G–I*). In an effort to find conditions under which inactivation could be reversible, we explored the effect of ATP. Presence of intracellular ATP interferes with calcium-induced desensitization, presumably due to occlusion of calcium-calmodulin binding at the N-terminal domain (*Lishko et al., 2007*; *Rosenbaum et al., 2004*). 10 mM ATP applied to the inside of patches had no effect on temperature activation and more importantly, on inactivation induced by repetitive activation (*Figure 3—figure supplement 1*).

## Inactivation impedes activation by capsaicin

To determine if current loss represents an inability of the channel to be reactivated by temperature or a widespread phenomenon, we examined capsaicin activation after channel inactivation. Surprisingly, activation by capsaicin is also impaired after multiple temperature ramps (*Figure 4*). Capsaicin elicits robust currents, but the response to a second application after several activating ramps, is highly diminished (*Figure 4A*). There is a clear inverse correlation between the extent of current loss induced by temperature and the magnitude of current activated by the second application of capsaicin (*Figure 4B*), suggesting that the remaining capsaicin-sensitive channels have not yet undergone inactivation. Application of the temperature ramp in the presence of capsaicin, produced an increase in current, as expected (*Figure 4C*), which had a reduced slope, since the channels are being activated by capsaicin and heat at the same time (*Figure 4D*). Remarkably, the channels entered the temperature-induced inactivated state regardless of the continued presence of capsaicin and the current was reduced to leak levels.

Several regions in TRPV1 channels, including the N-terminal ankyrin domains, the transmembrane domains, the pore region and the C-terminus, have been implicated in temperature gating (*Cui et al., 2012*; *Grandl et al., 2008*; *Yao et al., 2011*; *Laursen et al., 2016*; *Yang et al., 2010*; *Grandl et al., 2010*; *Zhang et al., 2018*). This apparently distributed heat sensitivity suggests that several regions contribute discretely to heat absorption or its coupling to channel opening. Here, we report on the existence of an inactivation process in TRPV1, which had not been previously characterized. Inactivation proceeds with a time course in the order of tens of seconds. The time course is strongly dependent on the maximum temperature attained during the activation ramp, behaving as if inactivation was coupled to opening. In this regard, temperature-dependent inactivation in TRPV1 resembles open state inactivation in voltage-dependent channels (*Demo and Yellen, 1991*; *Aldrich et al., 1983*). However TRPV1 inactivation is irreversible.

Our results are compatible with a unique sequence of events during heat absorption and its conversion to pore opening. Heat might be absorbed in several regions of the channel, leading to partial contributions to channel opening, but some regions might absorb heat through a mechanism akin to partial polypeptide denaturation. Channels undergoing this partial denaturation would become less able to absorb or convert heat into a conformational change, resulting in reduced apparent enthalpy of activation. Finally, after opening, channels become sufficiently distorted and enter the inactivated state irreversibly. This is consistent with the high heat capacity change ($\Delta C_p$) mechanism (*Clapham and Miller, 2011*; *Chowdhury et al., 2014*) and might reflect alterations of hydrophobic interactions through solvent exposure (*Sosa-Pagán et al., 2017*) and not large conformational changes (*De-la-Rosa et al., 2013*; *Ruigrok et al., 2017*).

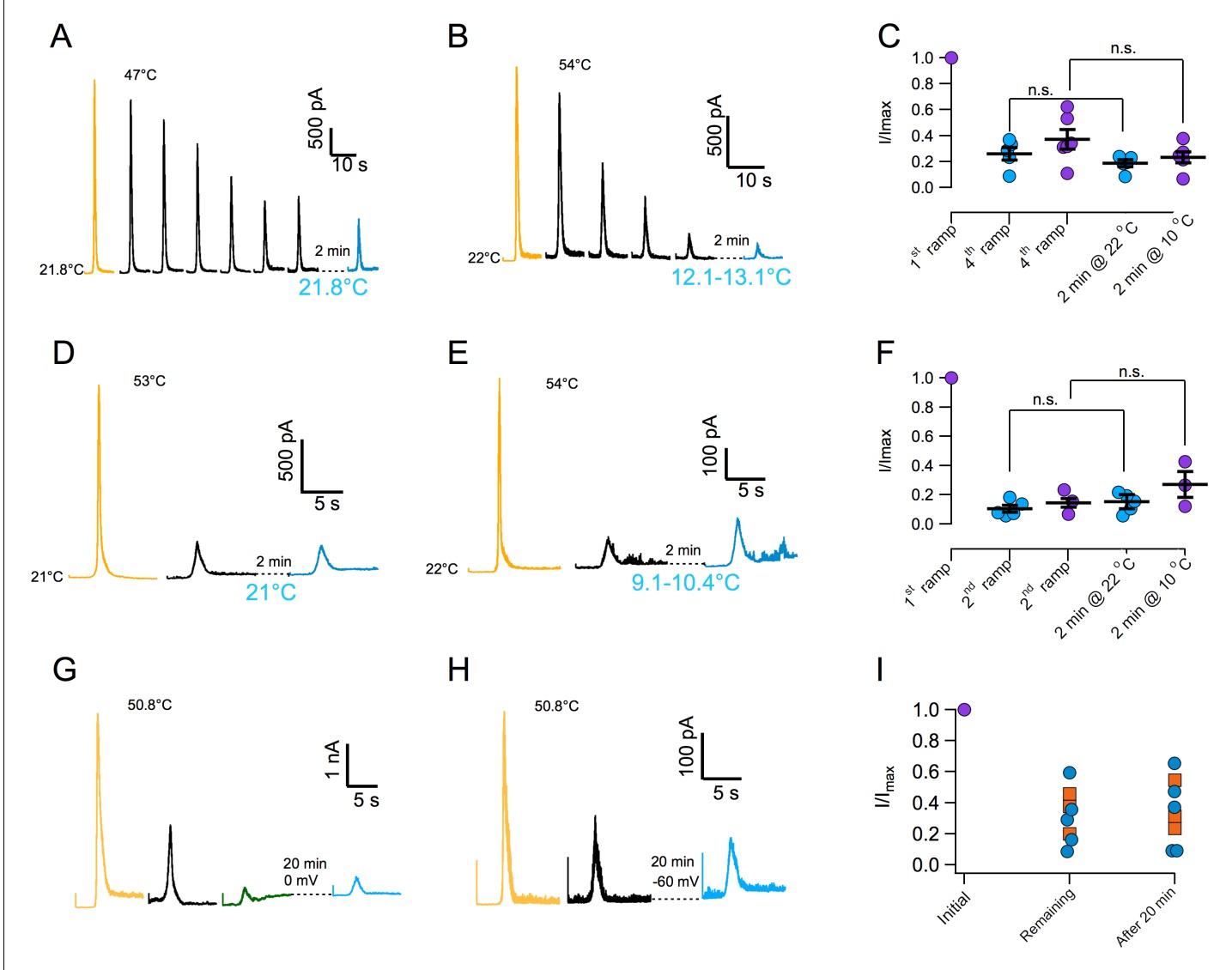

**Figure 3.** Inactivation of TRPV1 by multiple temperature ramps is irreversible. (A) In this particular patch, seven consecutive ramps to the temperature indicated were applied with an interval of 30 s to produce ~60% inactivation. The patch was left at the bath temperature (21.8°C) for a 2 min recovery period, after which another identical activation ramp did not elicit a current larger than the residual current after inactivation, indicating absence of recovery from inactivation. (B) Another patch was subjected to five activation ramps that produced ~80% inactivation. This time, during the 2 min recovery period, bath temperature was lowered to between 12 and 13°C. Again, no current was produced by an activation ramp, indicating no recovery. Maximal temperature during the ramp is indicated in each panel. (C) Summary of results from several experiments indicating no recovery of current after a 2 min period at normal bath temperature (~22°C, blue circles) or in a cooled bath (10°C, purple circles). Statistical difference was tested with a Student-t test, n.s. indicates no difference (p>0.2). (D) Similar to (A) but in this patch the current was elicited by a ramp to a larger temperature, indicated in each panel, which produced a larger inactivation in a single ramp (80%). After a 2 min period at 21°C, no current was recovered. (E) Same as in (C) but with a recovery period at 10°C. No recovery of current is evident. (F) Summary of results from several experiments as in (D) and (E) indicating no recovery of current after a 2 min period at normal bath temperature (~22°C, blue circles) or in a cooled bath (10°C, purple circles). Absence of significant recovery was based on a Student t-test with p>0.2. The label n.s. indicates non-significance. (G) No recovery of current is observed with a longer recovery period of 20 min at 0 mV or at −60 mV (H). (I) Summary of recovery of five patches held at 0 mV for 20 min after current inactivation (blue circles) and three patches held at −60 mV for 20 min after current inactivation (orange squares). No current is recovered, regardless of the fraction of inactivated current.

DOI: https://doi.org/10.7554/eLife.36372.009

The following source data and figure supplement are available for figure 3:

**Source data 1.** Raw data for *Figures 3C, 3F and 3I*.

DOI: https://doi.org/10.7554/eLife.36372.011

*Figure 3 continued on next page*

*Figure 3 continued*

**Figure supplement 1.** The presence of ATP fails to affect inactivation or produce recovery from inactivation.

DOI: https://doi.org/10.7554/eLife.36372.010

## Materials and methods

### Cell lines

We used the HEK293 cell line obtained from ATCC 10 years ago, cell identity has not been authenticated, but we have tested for mycoplasma and the cells are free of infection.

### Cell culture and transfection

HEK293 cells were grown on 100 mm culture dishes with 10 ml of DMEM, Dulbecco´s Modified Eagle Medium (Invitrogen) containing 10% fetal bovine serum (Invitrogen) and 100 Units/ml-100 µg/ml of penicillin-streptomycin (Invitrogen), incubated at 37˚C in a 5.2% $CO_2$ atmosphere. When cells reached 90% confluence, the medium was removed and cells were treated with 1 ml of 0.05% Trypsin-EDTA (Invitrogen) for 5 min. Subsequently, 1 ml of DMEM with 10% FBS was added. The cells were mechanically dislodged and reseeded in 30 mm culture dishes over 5 × 5 mm coverslips for electrophysiology or in 35 mm glass bottom dishes, for imaging. In both cases, 2 ml of complete medium were used. Cells at 70% confluence were transfected with the appropriate construct as indicated, using jetPEI transfection reagent (Polyplus Transfection). For patch-clamp experiments, pEGFP-N1 (BD Biosciences Clontech) was cotransfected with rTRPV1 to visualize successfully transfected cells via fluorescence. Electrophysiological recordings were done one or two days after transfection.

### DNA constructs

Electrophysiological experiments were carried out on rat TRPV1-WT cloned in the plasmid pcDNA3.

### Current recording

Patch clamp recordings were made from HEK293 cells expressing TRPV1 in the inside-out, whole-cell and outside-out configurations of the patch-clamp recording technique. Inside-out and outside-out recordings were made using symmetrical solutions consisting of 130 mM NaCl, 10 mM HEPES, 5 mM KCl and 5 mM EGTA for calcium-free conditions, pH 7.2 adjusted with NaOH. Whole-cell recordings of TRPV1 channels were made using a bath solution with 130 mM NaCl, 10 mM HEPES, 0.5 mM $CaCl_2$, pH 7.2 adjusted with NaOH and a pipette solution with 130 mM NaCl, 10 mM HEPES, 5 mM KCl and 5 mM EGTA.

Macroscopic currents were low pass filtered at 1 kHz, sampled at 20 kHz with an Axopatch 200B amplifier (Axon Instruments), acquired and analyzed with PatchMaster data acquisition software and using an Instrutech 1800 AD/DA board (HEKA Elektronik). Data acquisition was synchronized to the programmable power supply (Agilent) controlling the micro-heater using an Arduino Uno microcontroller. Pipettes for recording were pulled from borosilicate glass capillary and fire-polished to a resistance of 4–7 MΩ when filled with recording solution for inside- and outside-out recordings and 1–2 MΩ for whole-cell. Intracellular solutions in inside-out patches were changed using a custom built rapid solution changer. For whole-cell recordings all the bath solution was exchanged. Temperature ramps were applied while the membrane potential was held constant at the indicated value. For recovery experiments, the membrane potential was set at 0 mV or −60 mV during the recovery period to probe the effect of membrane potential.

4 mM capsaicin stocks were prepared in ethanol, stored at −20˚C and diluted to the desired concentration in recording solution as indicated before the experiments.

### Chemicals

All chemicals, including capsaicin and NaATP were purchased from Sigma-Aldrich.

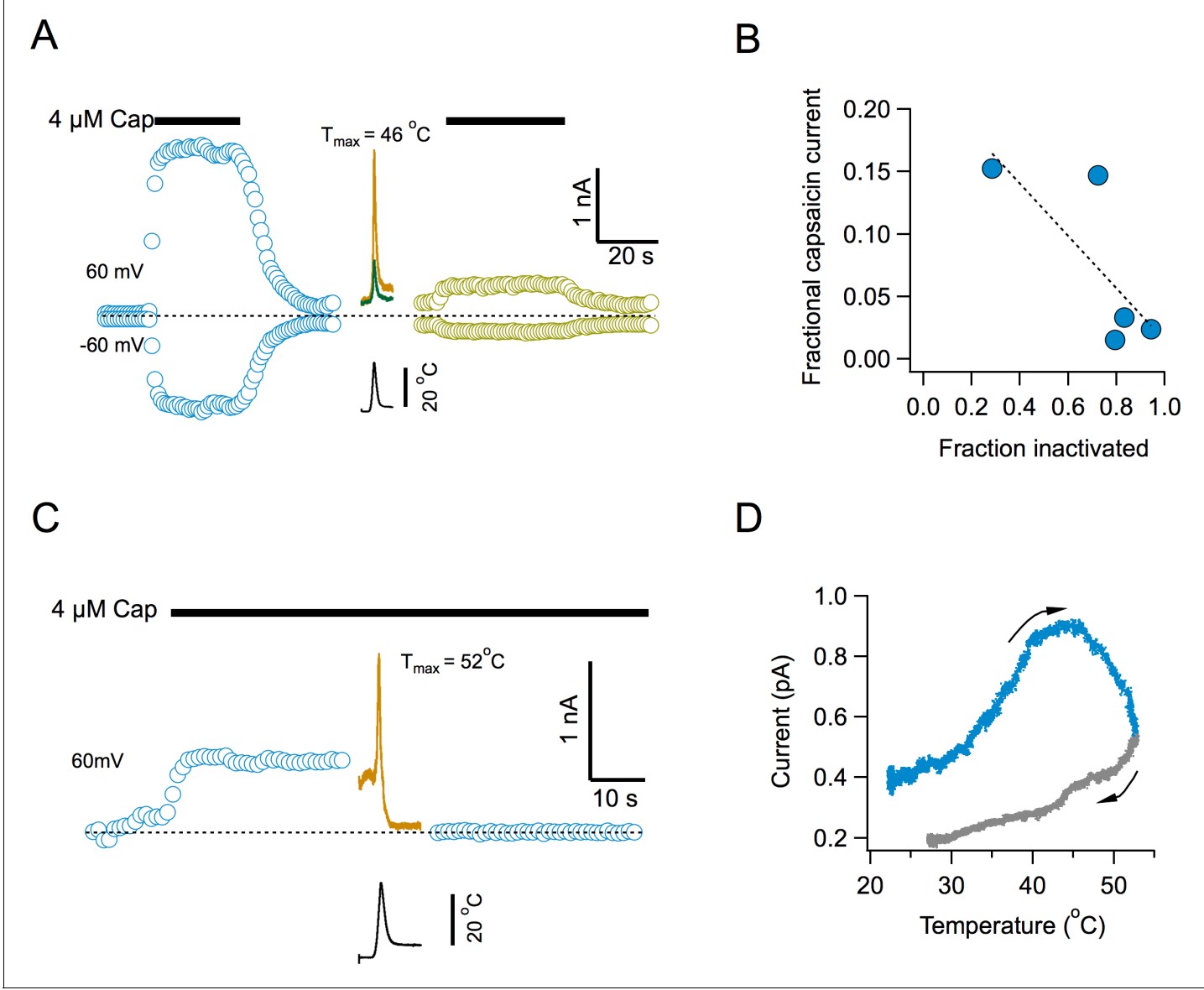

**Figure 4.** Inactivation by temperature renders the channel irresponsive to activation by capsaicin. (**A**) Response of channels in an inside-out patch to application of a saturating concentration of capsaicin before (blue) and after (lemon) application of 4 activation ramps to the indicated temperature (Middle traces, shown are the first and fourth temperature activation ramps, applied between the two applications of capsaicin). (**B**) Inverse correlation between the degree of inactivation of TRPV1 currents and the response to reapplication of 4 µM capsaicin. The dotted line is a fitted linear function through the data, with a correlation coefficient, r = −0.77. (**C**) Application of a temperature ramp while the channel is being activated by capsaicin causes loss of current. A saturating concentration of capsaicin was applied to an inside-out patch producing outward currents. A ramp to a high temperature that normally produces inactivation was applied in the continued presence of capsaicin. This ramp produces first an increase of current and an immediate reduction of current to leak levels, indicating current loss. (**D**) Plot of the current activated by the ramp as a function of temperature, indicating that current starts to decay even as the temperature is still increasing. The arrows indicate the direction of change of temperature.
DOI: https://doi.org/10.7554/eLife.36372.012

The following source data is available for figure 4:

**Source data 1.** Data for *Figure 4B*.
DOI: https://doi.org/10.7554/eLife.36372.013

## Temperature activation

Temperature activation was carried out using a micro-heater that uses the resistive heating principle as previously described (*Islas et al., 2015*), with the modification that, in this study, we used a programmable current-regulated DC power supply (Agilent) controlled via a Python program. For all recordings and temperature calibrations, the pipette with seal was placed approximately 27 μm from the micro-heater, in the case of whole-cell recording the cell was lifted and placed in front of the micro-heater. The membrane potential was held constant at the indicated potential, typically 60 mV, during temperature ramp application. Before experiments, the temperature reached by the micro-heater was calibrated by the relation between resistance and temperature in an open pipette (*Yao et al., 2009*; *Islas et al., 2015*), which was obtained from an open pipette submerged in a chamber homogeneously heated by a Peltier device and in which temperature was measured with a thermistor (Warner Instrument).

## Perfusion of solutions

For cooling experiments, the whole recording chamber was exchanged with a cooled bath solution. Cooling of the bath solution was achieved by surrounding the reservoir containing the solution in an ice bath with added sodium chloride. Bath temperature reached near the patch was measured with a thermistor.

For ATP experiments, the bath recording solution contained 10 mM ATP ($Na^+$ salt), so that inside out patches experience this ATP concentration throughout the recording.

## Data analysis

All data were analyzed and plotted using Igor Pro v6 (Wavemetrics, Inc.). Pooled data are presented with the mean and standard error of the mean (s.e.m.). Statistical significance was assessed with a Student's t-test as implemented in Igor pro. Significant differences between means were considered to exist went the p value was less than 0.01.

## Acknowledgements

We would like to thank Itzel Alejandra Llorente-Gil for expert technical help. This work was supported by grants from CONACYT CB-2015–252644 to LDI. and CB-2014-01-238399 to TR, CONACYT-Fronteras de la Ciencia 77 to TR and LDI, DGAPA-PAPIIT-UNAM IN209515 to LDI and IN200717 to TR.

## Additional information

### Competing interests

León D Islas: Reviewing editor, *eLife*. The other authors declare that no competing interests exist.

### Funding

| Funder | Grant reference number | Author |
| --- | --- | --- |
| Consejo Nacional de Ciencia y Tecnología | CB-2015-252644 | Leon D Islas |
| DGAPA-PAPIIT-UNAM | IN209515 | Leon D Islas |
| DGAPA-PAPIITT-UNAM | IN200717 | Tamara Rosenbaum |
| Consejo Nacional de Ciencia y Tecnología | CB-2014-01-238399 | Tamara Rosenbaum |
| Consejo Nacional de Ciencia y Tecnología | Fronteras de la Ciencia 77 | Tamara Rosenbaum Leon D Islas |

The funders had no role in study design, data collection and interpretation, or the decision to submit the work for publication.

## Author contributions
Ana Sánchez-Moreno, Formal analysis, Validation, Investigation, Writing—review and editing; Eduardo Guevara-Hernández, Formal analysis, Investigation, Writing—review and editing; Ricardo Contreras-Cervera, Resources, Investigation, Writing—review and editing; Gisela Rangel-Yescas, Resources, Data curation, Investigation; Ernesto Ladrón-de-Guevara, Software, Investigation, Methodology, Writing—review and editing; Tamara Rosenbaum, Resources, Funding acquisition, Investigation, Writing—original draft, Project administration, Writing—review and editing; León D Islas, Conceptualization, Data curation, Software, Formal analysis, Supervision, Funding acquisition, Validation, Methodology, Writing—original draft, Project administration

## Author ORCIDs
León D Islas https://orcid.org/0000-0002-7461-5214

## Decision letter and Author response
Decision letter https://doi.org/10.7554/eLife.36372.016
Author response https://doi.org/10.7554/eLife.36372.017

## Additional files

### Supplementary files
• Transparent reporting form
DOI: https://doi.org/10.7554/eLife.36372.014

### Data availability
Summary data for Figures 1, 2, 3, and 4 have been provided as source data files. The electrophysiological recordings will be made available upon request to the corresponding author.

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
