## [Decision Letter]

Thank you for submitting your article "Irreversible temperature gating in trpv1 sheds light on channel activation" for consideration by *eLife*. Your article has been favorably evaluated by Richard Aldrich (Senior Editor) and three reviewers, one of whom, Baron Chanda (Reviewer #1), is a member of our Board of Reviewing Editors.

The reviewers have discussed the reviews with one another and the Reviewing Editor has drafted this decision to help you prepare a revised submission.

Summary:

In this study, the authors present an interesting finding that has the potential to upend our current understanding of the thermodynamics underlying temperature-dependent gating of TRPV1 channels. Because of their exquisite heat sensitivity and physiological role as sensors of noxious temperature stimuli, these channels have become model systems to study the biophysical mechanisms of temperature-sensitivity. To date, the underlying assumption for any of the detailed thermodynamic analysis has been that the temperature-activation in TRPV1 channels is a reversible process and therefore can be modeled using widely used standard approaches. Here, Moreno et al. show that repeated activation of TRPV1 channels by thermal stimuli modifies the channel irreversibly such that the channel loses its ability to be activated either by heat or ligands such as capsaicin. These are remarkable findings that require us to rethink our current understanding of TRP channel gating for multiple reasons. Although the data is convincing and the main conclusions support their data, the reviewers have made specific recommendations.

Essential revisions:

1) Please reformat the manuscript for submission as a Short Report rather than a full-length article as proposed in the original decision letter. Much of the detailed modeling which is based on the assumption of equilibrium thermodynamics should be eliminated.

2) Please remove the DTNB experiments as the results of these experiments are not sufficiently conclusive and detract from the main findings.

3) Discus the Liu and Qin (2016) paper showing the reversibility of TRPV1 activation. It is preferable to show data that reconcile your findings with their studies. This would establish conditions under which reversibility of gating is a reasonable approximation.

4) Please add longer recovery times, different voltage protocols and other experiments that demonstrate the robustness of this phenomenon.

---

## [Author Response]

Essential revisions:1) Please reformat the manuscript for submission as a Short Report rather than a full-length article as proposed in the original decision letter. Much of the detailed modeling which is based on the assumption of equilibrium thermodynamics should be eliminated.

We have reformatted the manuscript as a Short Report and limited the number of main figures to four. We have removed all the modeling presented in the previous version of the manuscript.

2) Please remove the DTNB experiments as the results of these experiments are not sufficiently conclusive and detract from the main findings.

We have removed the DTNB experiments and the supplementary figure associated with it.

3) Discus the Liu and Qin (2016) paper showing the reversibility of TRPV1 activation. It is preferable to show data that reconcile your findings with their studies. This would establish conditions under which reversibility of gating is a reasonable approximation.

We now discuss the Liu and Qin (2016) paper. The published data in that paper regarding TRPV1 only shows the current in response to two temperature pulses. This makes it very hard to judge how stable the current in those experiments is. In fact, their Figure 9 shows a slight decrease in the magnitude of the current generated by the second pulse and intriguingly, also a small decrease in the enthalpy associated with the current elicited by the second pulse, which points in the same direction as the data presented here.

Trying to find conditions in which TRPV1 temperature–activated currents might be recoverable, we performed a new experiment in which intracellular ATP was present. ATP delays the onset of calcium induced desensitization and might have a similar effect on temperature-induced inactivation, however we find no effect on the rate of inactivation and recovery of current continues to be absent. These data are presented in Figure 3—figure supplement 1.

4) Please add longer recovery times, different voltage protocols and other experiments that demonstrate the robustness of this phenomenon.

We have performed and included results from experiments with a longer recovery time of up to 20 min. We performed these experiments with two different holding potentials during the recovery period: 0 mV, as we had done before for 2 min recovery, and -60 mV. These results show that there is no discernible recovery even for the longer times and that the holding potential has no effect on the recovery (Figure 3G-I).